# Analytical and Numerical Modeling of Degradation and Pyrolysis of Polyethylene: Measuring Aging with Thermogravimetry

**DOI:** 10.3390/polym14132709

**Published:** 2022-07-01

**Authors:** Tuukka Verho, Jukka Vaari

**Affiliations:** VTT Technical Research Centre of Finland Ltd., P.O. Box 1000, FI-02044 Espoo, Finland; tuukka.verho@vtt.fi

**Keywords:** cross-linked polyethylene, kinetic Monte Carlo, graph model, thermogravimetric analysis, aging

## Abstract

Aging reactions due to heat and radiation cause chain scissions and cross-linking in cross-linked polyethylene (XLPE). We have developed theoretical and numerical graph models to study the evolution of the gel fraction and network properties during aging as well as the mass loss during thermogravimetric analysis (TGA). Our analytical and kinetic Monte Carlo (KMC) based models that combine degradation reactions and a simple vaporization model can quantitatively predict TGA curves for aged XLPE. Fitting the model to experimental TGA data yields the number of scission reactions, showing that thermogravimetry combined with our models can present a nondestructive aging characterization tool for lifetime prediction.

## 1. Introduction

Polymer materials are subjected to various environmental stressors during their lifetime, which can lead to gradual degradation, or aging, of the materials [1,2,3]. Aging alters the physical properties of the polymer, such as elasticity, toughness, dielectric and combustion properties [4,5,6]. A particularly common aging mechanism is oxidative reactions due to elevated temperature or exposure to radiation. In most nonhalogenated polymers, oxidative degradation leads to changes in the chemical connectivity by chain scission and cross-linking reactions. These changes alter the network properties or molecular weight distribution (MWD) of the polymer, causing changes in crystallinity, density and mechanical properties. Typically, chain scission dominates over cross-linking, leading to a breakdown of cross-linked networks and progressive reduction in molecular weights.

Thermogravimetric analysis (TGA) is an experimental means to characterize the pyrolysis properties of a polymer. It is central to predicting the combustion properties of the material as it measures the generation of gaseous fuel during heating [7]. However, changes in the TGA curves due to oxidation are also indicative of the aging state of the polymer. An ability to quantitatively interpret these changes would allow the use of TGA—a straightforward experiment requiring only small material samples—to estimate the overall degradation of physical properties and the remaining service life of a polymeric component. Relatively few studies so far have used TGA in inert atmosphere to investigate the aging of polyethylenes. Wang et al. [8] studied thermal aging of low-density polyethylene (LDPE) cables at 100 °C and reported the appearance of a low-temperature decomposition process in highly aged samples. Weon [9] investigated thermal aging of linear low-density polyethylene (LLDPE) at 100 °C and found a shift in the TGA curves towards higher temperatures. We note that interpreting thermal aging results can be complicated by the fact that thermal aging may not only involve chemical changes but also evaporation of low molecular weight species, either present from the beginning [10,11,12] or produced by aging reactions [13]. Pablos et al. [14] studied room-temperature photodegradation of LDPE and LLDPE and reported earlier onset of mass loss in TGA for aged samples which were found to have roughly halved molecular weights compared to fresh ones. The reduction in the molecular mass of the polymer chains is a well-known effect of oxidative aging [15] and could explain the earlier onset of mass loss. In contrast, a radiochemical aging study in vacuum by Nyden et al. [16] reported a uniform 20 °C upward shift in temperature for the TGA curve and attributed it to anaerobic cross-linking. No study has attempted to provide a mechanistic explanation for the aging-related changes found in TGA curves.

Oxidative degradation of polymer often proceeds with random scission reactions, which generally means that many reactions are needed for the creation of volatile products. This precludes the use of lumped single-step reaction models to interpret TGA experiments on polymers [17]. However, with a knowledge of the reaction pathways for a given polymer, it is possible to solve systems of ordinary differential equations to predict the formation of various species during pyrolysis. Nevertheless, the huge number of distinct species in a disperse polymer system makes it a formidable task to keep track of all species and reactions. Nemeth et al. [18] modelled the degradation of a system of monodisperse short-chain PE by considering about 1000 species and 7500 elementary reactions. Levine and Broadbelt [19] used the method of moments to reduce the number of species, leading to a set of 151 species and 11,000 reactions. The computational effort required to solve such systems for realistic MWDs limits their use without simplifications in, e.g., computational fluid dynamics simulations [20]. The inclusion of chain branching and cross-linking will pose further challenges.

To overcome the complications posed by a plethora of different reactions and chemical species, a coarse-grained approach can be employed. In coarse-grained models, a small selection of repeat units is used to represent different chemical groups, and only a limited set of non-elementary reactions are considered. To simplify the solution of the kinetics, a reaction probability-based kinetic Monte Carlo (KMC) method can be used instead of solving systems of rate equations to track the chemical transitions due to the reactions [21,22]. Galina and Lechowicz [23] applied a Monte Carlo approach to polymer degradation and used coarse-grained network analysis to study the topological changes during aging. Adema et al. [24] used a coarse-grained KMC method to study the photodegradation of polyester-urethane coatings. Bystritskaya et al. [25] used KMC to study thermal degradation of polymers by the chain end monomer severing mechanism. Vinu et al. [26] used KMC to study thermal degradation of poly(styrene peroxide). KMC has also been used to study the cracking of alkanes [27] and heptane-4-suplhonic acid [28].

In this paper, we propose a coarse-grained graph representation of fresh and thermally degraded cross-linked polyethylene (XLPE) combined with analytical and numerical models to study the evolution of connectivity and volatile fraction during aging and pyrolysis. Disregarding local chemical changes and only considering topological evolution allows for studying very large systems and predicting network-dependent properties (mechanical etc.) and quantitatively predicting the TGA curves. Our analytical expressions allow rapid fitting to data while our numerical KMC models offer the flexibility to study more complicated systems and several types of reactions. We apply our models to identifying the aging states from experimental TGA results.

## 2. Theoretical Model

### 2.1. Analytical Model for Cross-Linked Polymer

Let us consider a polymer system with N monomers where the molecular weight follows the Flory–Schulz distribution, i.e., every monomer has the same probability to be a terminal unit. The chain length distribution density is given by
(1)$$P_{\mathrm{FS}}\left(k\right)=\frac{1}{\bar{k}}{\left(1-\frac{1}{\bar{k}}\right)}^{k-1}$$
where k is the chain length and with $\bar{k}=N/n_{c}$, $n_{c}$ being the number of chains. For unaged systems, the chains are sufficiently long so that the distribution can be approximated by the continuous exponential distribution,
(2)$$P\left(k\right)=\frac{1}{\bar{k}}\exp\left(-\frac{k}{\bar{k}}\right).$$

The dispersity of the distribution is D=2.

Introducing $n_{\mathrm{xl}}$ cross-links between the chains forms a polymer network that can be represented as a graph where the nodes are monomer units and edges are chemical bonds. The fraction of monomers in the macroscopic connected component is called the gel fraction g. Disregarding the possibility of monomers with more than one cross-link, the total number of monomers involved in a cross-link is $2n_{\mathrm{xl}}$.

To determine g after the gel has formed, consider removing and randomly reintroducing the cross-links of a randomly selected chain with length k. Every monomer has a probability of $f_{\mathrm{xl}}=2n_{\mathrm{xl}}/N$ to be a cross-link site, and the probability that the cross-link connects it to a gel monomer is g. Thus, the probability for a single monomer to be a cross-link site connected to the gel is $f_{\mathrm{xl}}g$, and the likelihood that the reintroduced cross-links do *not* connect the chain to the gel is ${\left(1-f_{\mathrm{xl}}g\right)}^{k}\approx \exp\left(-f_{\mathrm{xl}}gk\right)$, the exponential approximation being valid for sufficiently long chains and moderately cross-linked systems. 

As the removal and reintroduction of cross-links will not change the statistics of the system, the above likelihood should reflect the gel fraction of all chains of length k. The total gel fraction is the sum of the mass-weighed contribution from all chain lengths:(3)$$1-g=\int_{0}^{\infty }\frac{k}{{\bar{k}}^{2}}\exp\left(-\frac{k}{\bar{k}}\right)\exp\left(-f_{\mathrm{xl}}gk\right)dk$$
where $k/{\bar{k}}^{2}\exp\left(-k/\bar{k}\right)$ is the mass-weighed chain length distribution density. Evaluating the integral gives
(4)$$1-g={\left(\bar{k}f_{\mathrm{xl}}g+1\right)}^{-2}.$$

Solving for g gives
(5)$$g=\frac{\bar{k}f_{\mathrm{xl}}-2+\sqrt{{\bar{k}}^{2}f_{\mathrm{xl}}^{2}+4\bar{k}f_{\mathrm{xl}}}}{2\bar{k}f_{\mathrm{xl}}}.$$

We note that the likelihood of the chain having the length k and not being in the gel is proportional to $P\left(k\right)\exp\left(-f_{\mathrm{xl}}gk\right)$, which implies that the chain length distribution (ignoring cross-links) of chains outside the gel is also exponential, with a mean of
(6)$${\bar{k}}_{\mathrm{not}\;\mathrm{gel}}=\frac{\bar{k}}{\bar{k}f_{\mathrm{xl}}g+1}=\bar{k}\sqrt{1-g}$$
where we applied Equation (4). The number of chains in the gel is then
(7)$$n_{c,\mathrm{gel}}=n_{c}-\frac{\left(1-g\right)N}{{\bar{k}}_{\mathrm{not}\;\mathrm{gel}}}=n_{c}\;\left(1-\sqrt{1-g}\right).$$

To determine the number of cross-links that are in the gel, consider adding an additional cross-link to the system. The likelihood that neither monomer of the cross-link is in the gel is ${\left(1-g\right)}^{2}$. The number of cross-links in the gel is therefore
(8)$$n_{\mathrm{xl},\mathrm{gel}}=n_{\mathrm{xl}}\;\left[1-{\left(1-g\right)}^{2}\right].$$

Knowing these quantities, we can calculate the cycle rank of the gel, which is central to its elastic properties: $r=n_{\mathrm{xl},\mathrm{gel}}-n_{c,\mathrm{gel}}+1$. Relationships on polymer gels similar to the above have previously been presented by Charlesby [29].

### 2.2. Analytical Degradation Model

Aging leads to scission and cross-linking reactions in the system, both considered to occur randomly (here, we assume that the biasing effect of, e.g., crystal phases can be ignored). As the Flory–Schulz chain length distribution can be considered to result from random scissions occurring on a single huge chain, further random scissions (that occur at chains, not cross-links) will not change the character of the distribution, only its mean. Similarly, aging-related random cross-linking reactions (and scissions that affect cross-links) simply modify the number of cross-links in the system. As a result, a system of size N is completely determined by the number of chains and cross-links, irrespective of its state of aging. Suppose that aging involves chain scissions at a rate $R_{\mathrm{sc}}$ and cross-links at a rate $R_{\mathrm{xl}}$. The likelihood that a scission event occurs at an existing cross-link is $n_{\mathrm{xl}}/n_{b},$ where $n_{b}$ is the number of bonds in the system, $n_{b}=N-n_{c}+n_{\mathrm{xl}}$. We can write ordinary differential equations for the evolution of scissions and cross-links:(9)$$\frac{dn_{c}}{dt}=R_{\mathrm{sc}}\left(1-\frac{n_{\mathrm{xl}}}{n_{b}}\right)\;\frac{dn_{\mathrm{xl}}}{dt}=R_{\mathrm{xl}}-R_{\mathrm{sc}}\frac{n_{\mathrm{xl}}}{n_{b}}.$$

With initial conditions $n_{c}\left(0\right)=n_{c,0}$ and $n_{\mathrm{xl}}\left(0\right)=n_{\mathrm{xl},0}$, we obtain the solution:(10)$$n_{c}\left(t\right)=\left(n_{c,0}-N\right){\left(\frac{R_{b}}{n_{b,0}}t+1\right)}^{-\frac{R_{\mathrm{sc}}}{R_{b}}}+N\;n_{\mathrm{xl}}\left(t\right)=n_{c}\left(t\right)+R_{b}t-n_{c,0}+n_{\mathrm{xl},0}$$
where $R_{b}=R_{\mathrm{xl}}-R_{\mathrm{sc}}$ and $n_{b,0}=N-n_{c,0}+n_{\mathrm{xl},0}$. The above solution applies for $R_{b}\neq 0$; for $R_{b}=0$ we obtain
(11)$$n_{\mathrm{xl}}\left(t\right)=-\frac{R_{\mathrm{xl}}}{R_{\mathrm{sc}}n_{b,0}}+\exp\left(-\frac{R_{\mathrm{sc}}}{n_{b,0}}t\right)\;n_{c}\left(t\right)=n_{\mathrm{xl}}\left(t\right)-n_{\mathrm{xl},0}+n_{c,0}.$$

By normalizing by volume, we can obtain the more convenient intensive quantities $\rho_{\mathrm{xl}}=n_{\mathrm{xl}}/V$, $\rho_{\mathrm{sc}}=n_{\mathrm{sc}}/V$, $r_{\mathrm{xl}}=R_{\mathrm{xl}}/V,$ and $r_{\mathrm{sc}}=R_{\mathrm{sc}}/V$.

### 2.3. Analytical Expression for Volatile Fraction

TGA curves for polymers with a given $\bar{k}$ and $\rho_{\mathrm{xl}}$ can be predicted by calculating the degraded state as a function of temperature and assuming that all volatile species are lost from the sample mass. As TGA involves heavy degradation, it is necessary to use the accurate geometric distribution (Equation (1)) instead of the approximate exponential distribution for the chain lengths. For uncross-linked systems, it is fairly straightforward to calculate the mass fraction of chains of length up to an evaporation limit n by
(12)$$F_{n}=\overset{n}{\underset{k=1}{\sum }}\frac{k}{{\bar{k}}^{2}}q^{k-1}=1-\left(1+n-nq\right)q^{n}$$
where $q=1-1/\bar{k}$. With cross-links, the calculation becomes more challenging, as cross-links combine chains into larger molecules. However, a good approximation can be obtained by assuming that only dimers (pairs of chains linked by a cross-link) can be volatile, as larger aggregates are likely to be too heavy. We can thus approximate the volatile fraction as
(13)$$F_{n}=F_{n}^{\mathrm{unxl}}+F_{n}^{\mathrm{dimers}}.$$

We can calculate $F_{n}^{\mathrm{unxl}}$ in a similar fashion to Equation (12):(14)$$F_{n}^{\mathrm{unxl}}=\overset{n}{\underset{k=1}{\sum }}\frac{k}{{\bar{k}}^{2}}q^{k-1}q_{\mathrm{xl}}^{k}=\frac{q_{xl}}{{\bar{k}}^{2}{\left(1-q_{t}\right)}^{2}}\;\left[1-\left(1+n-nq_{t}\right)q_{t}^{n}\right]$$
where $q_{\mathrm{xl}}=1-f_{xl}$ and $q_{t}=qq_{\mathrm{xl}}$. As dimers are pairs of chains that have only one cross-link, we should calculate the mass fraction of such chains. By noting that the likelihood for a chain to have exactly one cross-link is $k\left(1-q_{\mathrm{xl}}\right)q_{\mathrm{xl}}^{k-1}$ according to the binomial distribution, we can write the mass fraction as
(15)$$F_{\infty }^{\mathrm{one}\;\mathrm{xl}}=\overset{\infty }{\underset{k=1}{\sum }}\frac{k^{2}}{{\bar{k}}^{2}}\left(1-q_{\mathrm{xl}}\right)q_{\mathrm{xl}}^{k-1}q^{k-1}=\frac{\left(1-q_{\mathrm{xl}}\right)\left(q_{t}+1\right)}{{\bar{k}}^{2}{\left(1-q_{t}\right)}^{3}}.$$

We can then obtain the mass fraction of dimers as follows: the probability for a randomly chosen monomer to be from a chain with one cross-link is $F_{\infty }^{\mathrm{one}\;\mathrm{xl}}$ and the probability that it is linked to another chain with one cross-link is $F_{\infty }^{\mathrm{one}\;\mathrm{xl}}/\left(1-F_{\infty }^{\mathrm{unxl}}\right)$. Therefore,
(16)$$F_{\infty }^{\mathrm{dimers}}=\frac{{\left(F_{\infty }^{\mathrm{one}\;\mathrm{xl}}\right)}^{2}}{1-F_{\infty }^{\mathrm{unxl}}}.$$

Finally, we need to resolve the volatile fraction among dimers. As the length distribution density of chains with a single cross-link is proportional to $kq_{t}^{k-1}$, we can calculate the mass-weighted size distribution for pairs of such chains by a convolution
(17)$$f_{k}^{\mathrm{dimer}}=\frac{k}{M}\overset{k-1}{\underset{j=1}{\sum }}jq_{t}{}^{j-1}\left(k-j\right)q_{t}^{k-j-1}=\frac{k\left(k-1\right)}{2}q_{t}^{k-2}{\left(1-q_{t}\right)}^{3}$$
where M signifies the normalizing constant to ensure that $\overset{\infty }{\underset{k=2}{\sum }}f_{k}^{\mathrm{dimer}}=1$, and the prefactor k weighs the distribution by mass. The volatile mass fraction of molecules with sizes up to n among dimers is
(18)$$\frac{F_{n}^{\mathrm{dimers}}}{F_{\infty }^{\mathrm{dimers}}}=\overset{n}{\underset{k=2}{\sum }}f_{k}^{\mathrm{dimer}}=1-\frac{Aq_{t}^{k}+Bq_{t}^{k+1}+Cq_{t}^{k+2}+Dq_{t}^{k+3}+Eq_{t}^{k+4}}{12q_{t}\left(q_{t}+1\right)}$$
where
(19)$$\begin{array}{l} A=n^{4}+4n^{3}+5n^{2}+2n\\ B=-4n^{4}-12n^{3}-2n^{2}+18n+12\\ C=6n^{4}+12n^{3}-12n^{2}-18n+12\\ D=-4n^{4}-4n^{3}+10n^{2}-2n\\ E=n^{4}-n^{2}. \end{array}$$

The volatile fraction of dimers $F_{n}^{\mathrm{dimers}}$ can be then calculated by multiplying Equation (18) by Equation (16). Combining the result with an expression for the temperature dependence for n allows to predict the mass loss in a TGA curve. Using n(T) given by Equation (24), Figure 1a shows that Equation (13) is in excellent agreement with a numerical simulation, while disregarding cross-links (Equation (12)) causes a shift in the curve toward lower temperatures. Figure 1b compares numerical results with Flory–Schulz and lognormal MWDs of the base polymer. A larger dispersity mainly has the effect of earlier onset of mass loss due to a larger initial number of short chains. As this effect is relatively minor, assuming a Flory–Schulz distribution should not cause a large error even if the real distribution has a different shape.

### 2.4. Coarse-Grained Kinetics of Pyrolysis

β-scission is the dominating molecular weight altering reaction in pyrolysis of polyethylene [18,19]. The reaction rate for β-scission is given by
(20)$$r_{\beta s}=\left[R\right]A_{\beta s}\exp\left(-\frac{E_{\beta s}}{RT}\right)$$
where [R] is the concentration of alkyl radicals and $A_{\beta s}$ and $E_{\beta s}$ are the Arrhenius constants. The concentration of alkyl radicals is given by a balance of homolytic fusion and radical recombination (Arrhenius constants denoted with subscripts *h* and *rc*, respectively):(21)$$\frac{d}{dt}\left[R\right]=2\left[C-C\right]A_{h}\exp\left(-\frac{E_{h}}{RT}\right)-2{\left[R\right]}^{2}A_{\mathrm{rc}}\exp\left(-\frac{E_{\mathrm{rc}}}{RT}\right)$$
where [C−C] is the concentration of C-C bonds. At elevated temperatures (close to 400 °C), the radical concentration equilibrates within a few seconds, and the left side of Equation (21) vanishes. We can then solve for [R] and substitute in Equation (20) to obtain
(22)$$r_{\beta s}=\left[C-C\right]A_{\mathrm{eff}}\exp\left(-\frac{E_{\mathrm{eff}}}{RT}\right)$$
where
(23)$$A_{\mathrm{eff}}=\sqrt{\frac{A_{h}}{A_{\mathrm{rc}}\left[C-C\right]}}A_{\beta s}\;E_{\mathrm{eff}}=E_{\beta s}+\frac{E_{h}-E_{\mathrm{rc}}}{2}.$$

As the reduction in the number of bonds during pyrolysis is rather small, we can with reasonable accuracy consider the concentration of C-C bonds as equal to the monomer concentration, [C−C]≈65 mol/l. Using the parameters in the modeling work by Levine and Broadbelt [19] in Equation (23) gives $A_{\mathrm{eff}}=2.00\times 10^{16}\;s^{-1}$ and $E_{\mathrm{eff}}=304\;\mathrm{kJ}/\mathrm{mol}$. The parameters by Nemeth et al. [18] give $A_{\mathrm{eff}}=5.15\times 10^{15}\;s^{-1}$ and $E_{\mathrm{eff}}=281\;\mathrm{kJ}/\mathrm{mol}$.

## 3. Materials and Methods

### 3.1. TGA Experiments with Aged XLPE

The XLPE material was obtained from Nexans NRC. The raw material was linear low-density polyethylene, and it was cross-linked at 65 °C using a silane cross-linker. The density and gel fraction of the resulting Si-XLPE were about 0.912 g/cm^3^ and 0.71, respectively. Thermo-radiative aging was performed in the Panoza facility at UJV Rez, Czech Republic, with a ^60^Co γ-ray source. The average temperature was 47 °C, and the average dose rate was 77.8 Gy/h. TGA experiments were conducted at VTT under a nitrogen atmosphere using a heating rate of 10 K/min. Char yield was checked by switching to air atmosphere and heating the crucibles up to 1000 °C.

### 3.2. Numerical Aging and Decomposition Model

Polyethylene is modeled as an undirected graph where vertices correspond to the CH_x_ monomer units and the edges are the chemical bonds between them. Here, only one vertex and edge type is considered, i.e., the effect of different local chemistries and bonds is disregarded. The structure of the network is stored in an adjacency list, which contains the bonded neighbors of each monomer. Random scission is performed by removing a random edge from the graph, and conversely, random cross-linking is performed by adding a new edge between two random vertices. The gel fraction is computed by identifying the largest connected component in the graph and counting the fraction of vertices contained by it. The cycle rank of the gel is calculated as the difference between the number of edges and number of vertices in the gel.

A standard KMC scheme is coupled to the graph model by assigning temperature-dependent per-bond scission rate constants to the graph edges. While a simpler approach would suffice for the present study with only one type of bond, this allows introducing multiple bond types in future work. The reaction rates follow Arrhenius kinetics defined by a pre-exponential factor A and activation energy E. Upon simulating a TGA experiment, a heating ramp is chosen and the simulation proceeds by choosing a bond to be scissed and calculating the length of the next time step. The probability of a bond being picked is proportional to its reaction rate, and the length of the next time step is taken from the exponential distribution using the sum of the reaction rates of all bonds as the rate constant. At a given temperature, molecules with n(T) (see below) or less carbon atoms are considered volatile and constitute the predicted mass loss. See the Data Availability Statement for the program codes.

### 3.3. Evaporation Limit

The evaporation limit n(T) in analytic and numerical calculations is found by fitting the Hill equation to the boiling point data T(n) of linear alkanes [30] and solving for n(T):(24)$$n\left(T\right)=\frac{w_{3}}{{\left(\frac{w_{1}-w_{0}}{T-w_{0}}-1\right)}^{1/w_{2}}}$$
where the fitting coefficients are $w_{0}=67.328$, $w_{1}=1191.8$, $w_{2}=0.90918$ and $w_{3}=20.941$.

## 4. Results

### 4.1. Model for Unaged XLPE

To construct numerical graph models of unaged XLPE, we first created linear polyethylene with Flory–Schulz molecular weight distribution. This was performed by starting from a single chain containing all the monomers and randomly cutting its bonds breaking it into chains of varying length. As expected, the average dispersity approaches the value 2, as shown in Figure 2a. However, after a sufficient number of scissions, the dispersity starts to show signs of decreasing. This indicates that the number of monomers per chain becomes low (under 20) and the statistics can no longer be approximated by the exponential distribution given by Equation (2). Snapshots at different number of scissions can be chosen to represent systems with different average molecular weights. Averaged molecular weight distributions are shown for two cases in Figure 2b. The numerical distributions follow the characteristic shape of the Flory–Schulz distribution, the position of the peak depending on $M_{w}$.

Figure 3 shows the evolution of gel fraction upon adding cross-links. The numerical results are in good agreement with the predictions of Equation (5). The gel appears when the number of cross-links per chain is $n_{xl}/n_{c}>0.25$. At 0.6 cross-links per chain, the gel fraction is ~0.71, a fairly typical experimental value for XLPE. The corresponding cross-link densities are 0.02 mol/L and 0.2 mol/L for the 56,000 g/mol and 5600 g/mol systems, respectively. In the remaining of the paper, the symbol $M_{w,\mathrm{chains}}$ is used to denote the weight average molecular weight of the XLPE without considering cross-links ($M_{w}$ before cross-linking).

### 4.2. Aging of XLPE

Having established numerical graph models for fresh XLPE, we proceeded to study their aging behavior. We modeled aging reactions in a coarse-grained way by introducing cross-links and removing bonds between monomers. In aging of polyethylenes exposed to oxygen, chain scissions typically dominate over cross-linking [15,31,32]. However, radiochemical aging in an inert atmosphere leads to the dominance of cross-linking [33]. We subjected our graph models to aging at different ratios of scission and cross-link reactions and compared with analytical predictions from Equation (5), as shown by Figure 4.

When scissions dominate over cross-linking, the network gradually breaks down, as expressed by the diminishing gel fraction. With low amounts of cross-linking ($p_{\mathrm{xl}}\leq 0.2$), the gel fraction decays rather quickly, although the lifetime of the gel is prolonged by any cross-linking present. The cycle rank follows a similar trend as the gel fraction, indicating loss of mechanical properties. However, with higher amounts of cross-linking ($p_{\mathrm{xl}}=0.33$), the decrease in the gel fraction slows down. This shift reflects the evolution of the system towards more random network structure, where a lower total number of bonds is sufficient to form a gel. This is also reflected in the growing number of cycles, which indicates mechanical stiffening of the polymer. Eventually, loss of the gel and mechanical properties is inevitable as the total number of bonds keeps decreasing. Yet, with equal amounts of scission and cross-linking ($p_{\mathrm{xl}}=0.5$), no loss of gel is found as expected. The analytical model predicts the numerical outcomes well, except for very high numbers of reactions when the number of very short chains becomes significant. When chains become very short, the exponential approximation (Equation (2)) leads to an overestimation of gel persistence.

### 4.3. Aging Characterization by TGA

The thermogravimetric data for fresh and thermo-radiatively aged XLPE is presented in Figure 5a (solid lines). For the fresh material, the decomposition occurs in a single step in the 400–500 °C temperature range, with a peak degradation rate at 470 °C. In aged samples, a low-temperature tail develops in the curves at temperatures above 150 °C. This tail involves no distinct degradation processes, however, as the differential curves in Figure 5b only show one peak. For the longest aging duration of 210 days, the low-temperature tail includes about 30% of the total mass. No char yield is found after the degradation.

Figure 5a also shows the results of fitting the theoretical model presented in Section 2.3 to the data. The input parameters for the model are $M_{w,\mathrm{chains}}$, $\rho_{\mathrm{xl}}$ and the effective Arrhenius parameters *A* and *E* for bond scission (see Equation (21)). A common value of *A* and *E* was fitted for all aged samples. For the unaged sample, different kinetic parameters were allowed, as using the same parameters as for aged samples led to a poor fit and an unphysically large $M_{w,\mathrm{chains}}$. A likely reason for the discrepancy is the presence of an undisclosed but small amount of storage antioxidant in the XLPE, according to information from the manufacturer. Several researchers have reported a shift in the TGA curve under N_2_ atmosphere to higher temperatures due to antioxidants [34,35,36]. To reduce the number of fitting parameters, cross-link densities were inferred from known data and initial fitting results. For the unaged sample, a cross-link density of 0.195 mol/l was calculated using Equation (4) based on an initially fitted $M_{w,\mathrm{chains}}\approx 5640\;g/\mathrm{mol}$ and the known gel fraction of 71% [32]. For the aged samples, we calculated the number of cross-links, assuming $p_{\mathrm{xl}}=0.2$. Judging from the elongation at break values in Hettal et al. [32], the gel is likely still present after 42 days of aging. Their kinetic model can be used to estimate the number of scissions at 42 days, giving 0.91 mol/l. From Figure 4, it can be seen that if there were no simultaneous cross-linking reactions ($p_{\mathrm{xl}}=0$), the gel would be completely lost after 42 days. Assuming $p_{\mathrm{xl}}=0.2$ gives a 64% reduction in cycle count for the sample aged for 42 days and 82% reduction after 84 days, which seems roughly appropriate.

After establishing the cross-link densities, we calculated updated values for $M_{w,\mathrm{chains}}$. The results of the fit are presented in Table 1, together with the rate constant *r_sc_* at 450 °C for bond dissociation, cross-link density, and fraction of bonds scissed $S=\left(n_{b}-n_{b,0}\right)/n_{b,0}$. Our kinetic parameters are of the same order as the theoretical estimates based on literature data presented in Section 2.4 and comfortably inside the rather wide range of reported literature values for pyrolysis of various polyethylene grades [37,38]. Finally, we note that the model curves were calculated considering 150 °C as the initial temperature, as the experimental data showed mass loss only after 150 °C, indicating the absence of species that would vaporize below 150 °C. As such species would be expected as a result of the aging reactions, we believe that the slightly elevated radiochemical aging temperature (47 °C) was sufficiently high to slowly evaporate anything that would vaporize below 150 °C in TGA.

Figure 6 compares the fraction of scissed bonds to the predictions of an analytical aging model (Equation (61) in [32]) developed for the same XLPE material and aging conditions. Both models predict a linear dependence of chain scissions on aging time, which is expected as the main initiating reaction is radical generation due to a constant radiative dose rate. Our thermogravimetry-based estimate for the scission rate (0.016%/d) is somewhat lower than the rate predicted by the kinetic model of Hettal et al. [32] (0.029%/d).

## 5. Discussion

In this work, we developed two methodologies to model the aging and pyrolysis of polyethylene: analytically derived equations and coarse-grained numerical models. The strength of the analytical equations is that they can be rapidly evaluated to explore the parameter space and fit to experimental data. Furthermore, they provide more high-level understanding than numerical simulations. However, a number of simplifying assumptions are necessary in the derivations, which prevents the study of different molecular weight distributions, branched polymers and different types of reactions. Numerical simulations, in turn, offer more flexibility and allow the study of more complicated systems. Therefore, the analytical and numerical approaches presented here complement each other to enable a wide range of applications.

Our aging models show that cross-linking reactions during aging can compensate for chain scissions. A combination of scissions and cross-link reactions lead to a more random molecular network topology that forms a gel and a large number of cycles with a lower total number of bonds. Therefore, cross-linking does not need to constitute more than a third of the reactions in order to theoretically avoid breakage of the gel for a very long time. However, at very large numbers of aging reactions, accurate predictions should take the chemical changes due to oxidation into account when calculating reaction rates.

Our results suggest that roughly 20% of all aging reactions should be cross-linking in order to explain slower than expected deterioration of the gel in Hettal et al. [32]. While this seems plausible in the light of the results by Fayolle et al. [15], where a similar fraction was found in thermal aging of polyethylene, there is also a possibility that this is an artifact of some simplification in our model. In particular, we ignored the effect of crystallinity on the distribution of aging reactions. In reality, aging reactions are concentrated in the amorphous phase, which has a high fraction of short chains.

Our results show that TGA combined with analytical or numerical degradation models could be a powerful aging characterization tool for a variety of applications. However, the degradation pathways for each polymer in question should be well understood in order to construct an accurate model. Furthermore, any fillers and additives that can evaporate or affect the oxidation kinetics in TGA will cause additional contributions in the signal that must be accounted for. Further research is thus needed to assess the potential of the method for specific applications.

While we have highlighted the application of our pyrolysis model on aging characterization, it should also be noted that it can be used to reveal hidden features of cross-linked polymer networks. Conventionally, it is difficult to determine the chain length distribution in cross-linked systems because the cross-links connect the chains to form a macroscopic molecule. However, our model separates the roles of chain backbone bonds and cross-links, and thus has the ability to uncover the pre-crosslinking $M_{w}$ of the polymers from a TGA measurement.

## 6. Conclusions

We developed analytical and numerical models for aging and pyrolysis of XLPE. The analytical models allow rapid evaluation and fitting to experimental data, while the numerical models enable the introduction of more complexity and freedom to define arbitrary molecular weight distributions. We found that, if present, cross-linking reactions can play a decisive role in aging of XLPE by mitigating the breakdown of the gel and reduction in network cycle count. We showed that our pyrolysis models can quantitatively reproduce the TGA curves of fresh and aged XLPE. By fitting our model to TGA data, the average chain length (as would be measured without cross-links) and number of scission reactions in aged samples can be elucidated. We compared our results for scission reaction rates during thermo-radiative aging to a previous result based on kinetic modeling and obtained a somewhat lower estimate.

## Figures and Tables

**Figure 1 polymers-14-02709-f001:**
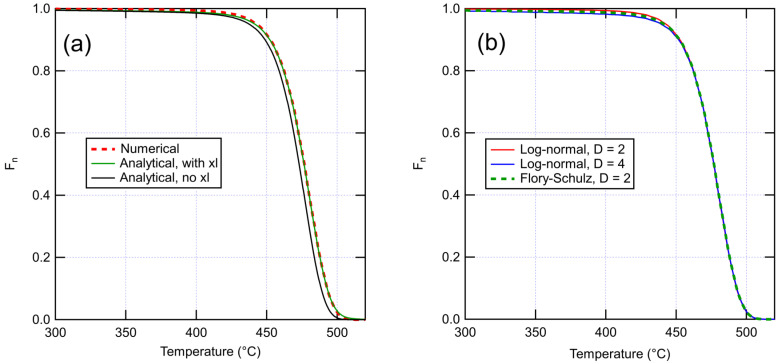
(**a**) Comparison of numerical and analytical TGA curves. Numerical curve is for a system of 10^5^ monomers with 700 chains and 615 cross-links (corresponding to $M_{w}=4000\;g/\mathrm{mol}$, $\rho_{\mathrm{xl}}=0.4$ mol/l). Analytical curves are for the same system with and without crosslinks. (**b**) Effect of dispersity on the numerical TGA curves. Results are shown for dispersity of 2 sampled from both log-normal and Flory–Schulz distributions with the same $M_{w}$, and dispersity of 4 sampled from log-normal distribution. Here, the scission rate $r_{\mathrm{sc}}$ is taken to follow an Arrhenius law with the constants $A=1.00\times 10^{14}\;s^{-1}$ and E=248 kJ/mol. Numerical results are averaged over 100 runs.

**Figure 2 polymers-14-02709-f002:**
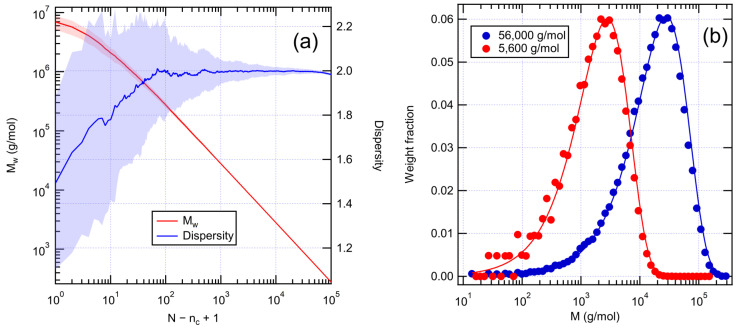
(**a**) Weight average molecular weight and dispersity of the system as a function of the number of random bond scissions applied to an initial single chain consisting of 10^6^ monomers. The results are averaged over 100 runs. Shading represents standard deviation. (**b**) Initial molecular weight distributions as averages over 100 runs. Solid lines are Flory–Schulz distributions for 0.05% (56,000 g/mol) and 0.5% (5600 g/mol) of initial bonds scissed.

**Figure 3 polymers-14-02709-f003:**
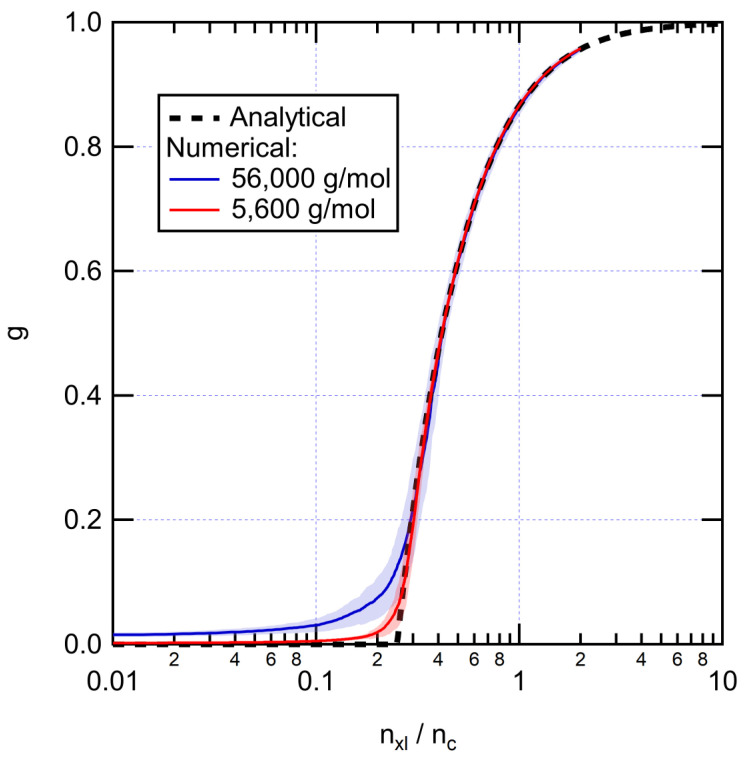
Evolution of the gel fraction as a function of the number of cross-links per chain for initially linear PE systems. Numerical results are averaged over 100 runs. Shading represents standard deviation.

**Figure 4 polymers-14-02709-f004:**
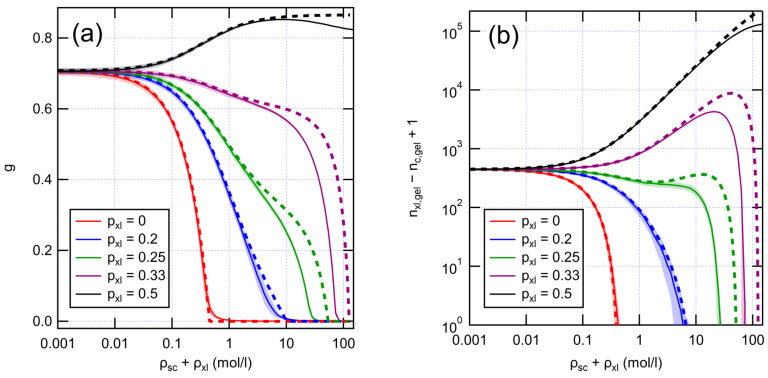
Evolution of (**a**) gel fraction and (**b**) cycle rank as a function of aging reactions with different values for $p_{\mathrm{xl}}$. Initial XLPE has $M_{w,\mathrm{chains}}=5600$ g/mol and the total number of monomers is 10^6^. Solid and dashed lines are results from numerical and analytical models, respectively. Numerical results are averaged over 100 runs. Shading represents standard deviation.

**Figure 5 polymers-14-02709-f005:**
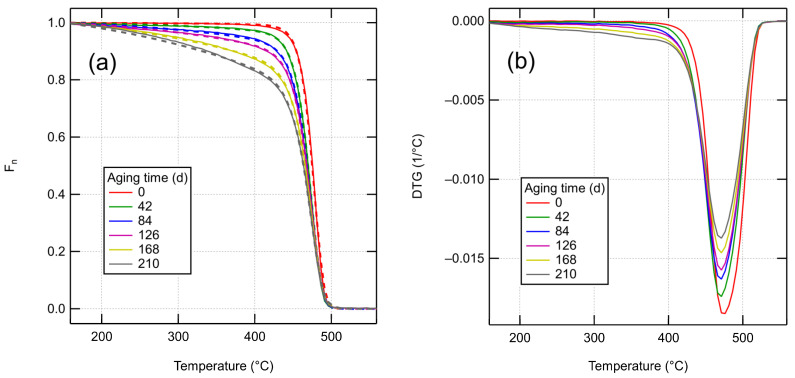
TGA curves. (**a**) Solid lines show experimental TGA curves, while dashed lines show the fitted model curves. (**b**) Differentiated experimental TGA curves.

**Figure 6 polymers-14-02709-f006:**
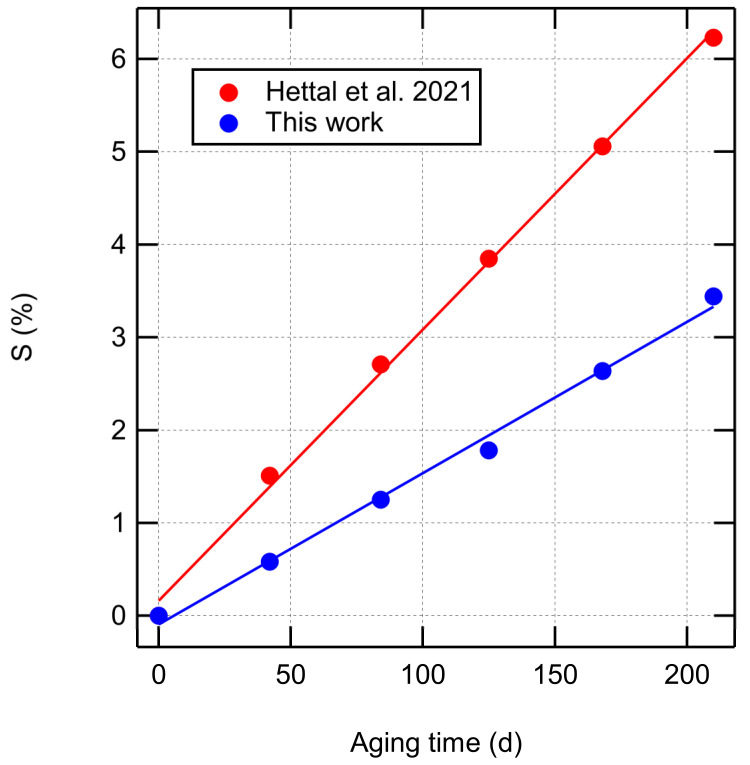
Fraction of scissed bonds as a function of aging time from Table 1 compared to the prediction of an analytical aging model [32].

**Table 1 polymers-14-02709-t001:** Results of fitting the theoretical model to the experimental TGA data.

Aging Time d	*M*_w,chains_ g/mol	*A* 1/s	*E* kJ/mol	*r_sc_* @ 450 °C mol/L/s	*ρ*_xl_ mol/L	*S*
0	5630	1.00 × 10^16^	276	1.09 × 10^−4^	0.195	0
42	2390	1.16 × 10^15^	262	1.45 × 10^−4^	0.283	0.0067
84	1500	1.16 × 10^15^	262	1.45 × 10^−4^	0.372	0.0137
126	1220	1.16 × 10^15^	262	1.45 × 10^−4^	0.460	0.0180
168	903	1.16 × 10^15^	262	1.45 × 10^−4^	0.548	0.0260
210	724	1.16 × 10^15^	262	1.45 × 10^−4^	0.637	0.0337

## Data Availability

Simulation codes developed for this work can be found at https://github.com/tverho/polymer-degradation (accessed on 21 June 2022).

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
