# Peer review of "Analytical and Numerical Modeling of Degradation and Pyrolysis of Polyethylene: Measuring Aging with Thermogravimetry"

_polymers, 2022, doi:10.3390/polym14132709_

Round 1

Reviewer 1 Report

This a nice, well-written paper. The work has many mathematical expressions, and many variables are defined through it, therefore sometimes it is hard to read it.

I have some minor comments that I think may help the reader to follow the paper:

·        I suggest, for clarity, to use the in the figures the variable defined in the equations. For example, in Figure 1 in the y-axis is written Mass (%) and I think that is the mass fractions Fn defined by eq 12 and 13. Also it is better that the x-axis in these curves cover the same range of temperatures.

·        Conversely, molecular weight average and cross-links density appear in line 143 and in figure 1 are named Mw and pxl (it is not clear if the cross-link density is p or rho)

·        Mw sometimes is written in Kg/mol and sometimes in g/mol

·        Also Aeff and Eff are used in figure 1 before they are defined in equation 23. These values are different from the ones of lines 200 and 201.

·        Figure 2 needs more explanation. Which is scissions? And initial bonds scissed? Which is the conclusion of Figure 2(b)?

·        In line 262, cross-link per chain is the same that cross-link?

·        In the caption of figure 5 appears, without definition in the text, Mw,chains.

·        The explanation in lines 327-329 about aged samples is not very clear

·        The same comment that, in general is to clarify and unify the variables written in the equations and shown in the figures, can be said about Table 1: S? r@450? Etc.

·        In line 340 is correct a value of 47ºC for the temperature?

·        In point 5 Discussion, from which previous result it is concluded that roughly 20% of all aging reactions should be cross-linking?

Reviewer 2 Report

This paper deals with the degradation and pyrolysis of polyethylene. A theoretical and numerical approch is proposed and discussed. The paper is well written and I was totally convinced by the results. Actually this research is of importance for polymer recycling. In the conclusion, the authors should extent their discussion to copolymer such as EPDM and how their results could be useful for more industrial polymers in rubber indutry. It is just a suggestion! Actually, it is a more general comment because  most of polymers are formulated. Polyethylene can contain fillers, fire retardants, plasticizers or other organic compounds. How this modeling and simulation can be extent to these complex industrial systems?
